# Phytotoxic Activity of *p*-Cresol, 2-Phenylethanol and 3-Phenyl-1-Propanol, Phenolic Compounds Present in *Cistus ladanifer* L.

**DOI:** 10.3390/plants10061136

**Published:** 2021-06-03

**Authors:** Cristina Tena, Ana del Rosario Santiago, Dolores Osuna, Teresa Sosa

**Affiliations:** 1Department of Plant Biology, Ecology and Earth Sciences, Faculty of Science, University of Extremadura, 06006 Badajoz, Spain; ctenacor@alumnos.unex.es; 2Center for Scientific and Technological Research of Extremadura (CICYTEX), Department of Crop Protection, 06187 Badajoz, Spain; anarosario.santiago@juntaex.es (A.d.R.S.); mariadolores.osuna@juntaex.es (D.O.)

**Keywords:** phytotoxicity, phenolic compounds, allelopathy, interaction of phytotoxic compounds, *Cistus ladanifer*

## Abstract

Numerous studies about the leaf exudate of *Cistus ladanifer* highlight this Mediterranean shrub as an allelopathic species. Despite the very high diversity of secondary metabolites identified in its labdanum, only a few components have been evaluated. *p*-Cresol, 2-phenylethanol and 3-phenyl-1-propanol are three phenolic components present in the labdanum of *C. ladanifer* whose role has not been specified to date. The present study, through a static acute toxicity test, analyzed their activity, with respect to *Allium cepa* and *Lactuca sativa*. These three separate compounds and the mixture of all of them have a more or less phytotoxic activity depending on the medium, species and concentration tested. When the test is carried out on paper, the three pure allelochemicals and their mixture at 1 mM significantly inhibited the total germination, the germination rate and the development of the *Allium cepa* and *Lactuca sativa* seedlings to a lesser extent, but when the test performed in soil, the effects on the size of roots and cotyledons are attenuated. Furthermore, in the two species tested on paper, the joint action of the three compounds at 1 mM shows a significantly greater inhibition of the measured indices than each of the compounds separately.

## 1. Introduction

Allelopathy, according to the International Allelopathy Society (IAS), “studies any process that involves secondary metabolites produced by plants, algae, bacteria and fungi that influence the growth and development of agricultural and biological systems” [1]. This type of chemical interaction mediated by secondary metabolites affects plants at the individual, population, community and ecosystem levels. At the individual level, one plan can generate allelochemicals that inhibit the germination of seeds and/or the growth of seedlings around it. The consequence of this effect at the community level can be observed as a lower floristic diversity in communities where allelopathic species are present [2,3,4,5,6].

An example of this type of allelophatic species is *Cistus ladanifer* (“rockrose”), an aromatic shrub of the family Cistaceae typical of the Mediterranean ecosystem. This shrub is distributed throughout Southwestern Europe and Northern Africa, from Portugal and Morocco to Southern France and Algeria [7]. Previous studies about rockrose populations showed that the floristic diversity found in these communities is lower than that found in neighboring plots of land where *C. ladanifer* is not present [8,9]. Numerous studies indicate that this observation could be attributed to the allelopathic potential of this species [10,11]. *Cistus ladanifer* secretes a large amount of exudate (labdanum) in its leaves and photosynthetic stems. A total of 358 different compounds have been identified in this labdanum rich in secondary metabolites, including: phenols, terpenes, alkaloids, polyacetylenes, fatty acids and steroids [12,13,14,15,16,17,18,19]. Different phytotoxicity studies with aqueous leaf extract, exudate fractions and some of the identified compounds show the allelopathic potential of this species [20,21,22,23,24,25,26,27,28,29,30,31]. Moreover, studies on the phytotoxicity, bioavailability, persistence and incorporation of these compounds in the soil have helped to elucidate their actual participation in allelopathic interaction [32,33,34,35,36,37,38]. However, the literature on the allelopathic activity of *C. ladanifer* is not complete. Due to the difficulty of isolating its components in a sufficient quantity and the fact that they are not commercially available, many of the components of this species have not been evaluated to date, including several phenolic compounds [10,11].

Phenolic compounds are a very diverse group of secondary metabolites characterized for having one or more hydroxyl groups (-OH), acidic reaction and an aromatic ring (phenol group). These compounds have many and very varied biological functions, from structural components and substances that attract pollinators, to poisons, deterrents, antifungals and antimicrobials [39]. Phenolic compounds are widely known for being allelopathic. Some examples include syringic, caffeic and ferulic acids, which are water-soluble components that are rapidly released from the leaves and roots to the surrounding soil, where they act as inhibitors of germination [39]. Studies on sprout extracts of *C. ladanifer* with water, where the phenolic compounds were removed, show that neither the complete extract nor the extract without phenols affected the total germination or the germination rate of *Trifolium subterraneum*, whereas the phenolic compounds reduced germination significantly, thus supporting the hypothesis that different types of chemical substances can affect different aspects of germination in a different manner [10]. In the essential oil of *C. ladanifer*, there is a predominance of phenolic compounds and flavonoids, with a significant amount of *p*-cresol, 2-phenylethanol and 3-phenyl-1-propanol [40]. Low molecular weight phenolic compounds have also been found in all the delignification liquors obtained in the literature, with *p*-cresol being one of the most abundant, and, in DCM leaf extracts, 2-phenylethanol is one of the main components [41,42]. Despite being so abundant in *C. ladanifer*, the phytotoxic activity of these three phenolic compounds has not been evaluated to date.

The general aim of the present work is to contribute to the understanding of the allelopathic activity of *C. ladanifer*. To this end, it was fundamental to determine the activity of all the components of the exudate. Through different bibliographic reviews [10,11,43,44,45,46], it was observed that phenolic compounds such as *p*-cresol, 2-phenylethanol and 3-phenyl-1-propanol have not been evaluated to date, although their recent commercial availability allows assessing them these days. The specific aim was a bioassay conducted with a static acute toxicity test. In order to use representative models of the two taxonomic classes of plants (mono- and di-cotyledoneae), commercial seeds of *Lactuca sativa* and *Allium cepa* were selected. These species are ideal for adequately showing the allelochemical effects on the germination processes [18,47]. Moreover, *L. sativa* is recommended by the US EPA (United States Environmental Protection Agency) for phytotoxicity tests, being among the most sensitive species [48]. *Allium cepa* has proved to be the most useful, and has repeatedly been suggested as a standard test material by the International Organization for Standardization [49], and the Organization of Economical Cooperation and Development the citation in the list of species historically used in plant testing [50]. Furthermore, ecologic agriculture and sustainable processes are gaining importance and social acceptance, and they have been focused on obtaining products with minimum environmental impact. Therefore, in a broader scope, the additional aim of the present study was to identify promising sources of natural herbicides.

## 2. Results

### 2.1. Effect of Phenolic Compounds on Lactuca sativa and Allium cepa Germination

2-phenylethanol and 3-phenyl-1-propanol significantly affected the seed germination of *Allium cepa* at all the tested concentrations, both in soil and on paper. For *p*-cresol, no significant germination inhibition was found at 0.5 mM and 0.1 mM when the test was conducted on paper, although such inhibition was detected when soil was used (Table 1). Regarding the germination of *Lactuca sativa*, the effect of these compounds was smaller, with no negative effect when soil was used. Only 2-phenylethanol and *p*-cresol significantly inhibited germination on paper at 1 mM, and at 1 mM and 0.5 mM, respectively (Table 2). Moreover, *p*-cresol, 2-phenylethanol and 3-phenyl-1-propanol significantly reduced the germination rate of *Allium cepa* and *Lactuca sativa* in the two soils and at the three concentrations tested, except in *L. sativa* at 0.1 mM when paper was used.

When tested with the mixture of the three compounds, a significant inhibition of total germination and germination speed at 1 and 0.5 mM is observed in the two species and substrates tested, and also a significant inhibition of the two indices is observed at 0.1 mM in *A. cepa* in soil and %S in *L. sativa* in soil and paper. On the other hand, in the two species studied when the test was carried out on paper, the joint action of the three compounds shows a significantly greater inhibition of the total germination and germination speed than when they acted alone less than 0.1 mM in *L. sativa* where the differences were not significant in %TG, but in addition, the inhibitory effect on the %S of *A. cepa* dissipated with the mixture of compounds at 0.1 mM. Likewise, when the test was carried out in soil, the joint action of the three compounds shows a significantly greater inhibition of total germination and germination speed, but also with some exceptions. In this case, in *A. cepa* not significant differences were found between the pure compounds with the mixture in% TG at 1 mM and in% S at 0.5 and 0.1 mM (Table 1 and Table 2).

It is worth highlighting that the greatest inhibition of total germination and germination rate was obtained at the highest concentration tested (1 mM) in every case, regardless of the compound, mixture, species or paper/soil. Only when *Allium cepa* was watered with 2-phenylethanol in soil, the germination inhibition was greater at the lowest concentration (0.1 mM), but the difference with the other concentrations was not significant. In all the treatments on *A. cepa* that were carried out on paper, significant differences are observed in the %TG between concentrations of 1 and 0.5 mM. However, in soil a significant difference is only observed in *p*-cresol. Furthermore, on paper the effects of 3-phenyl-1-propanol and mixture of compounds showed a significant positive correlation with concentration (*r*^2^ = 0.98 and *r*^2^ = 0.99 respectively). For %S the same behavior is observed, we found significant differences between the concentrations of 1 and 0.5 mM in all the treatments that are carried out on paper, but in soil significant differences are only observed in *p*-cresol and a mixture of compounds. In paper, the effects of *p*-cresol, 3-phenyl-1-propanol and mixture of compounds on germination speed showed a significant positive correlation with concentration (*r*^2^ = 0.98, *r*^2^= 0.96 and *r*^2^ = 0.99 respectively), in soil the effect was lower, but significant differences were found between 0.5 and 1 mM for *p*-crepsol and the mixture of compounds (Table 1). In the different treatments on *Lactuca sativa* in %TG, significant differences between concentrations are only observed when it is treated with the mixture of compounds on paper, finding a significant positive correlation with concentration (*r*^2^ = 0.99). In %S, significant differences are observed between concentrations of 1 and 0.5 mM in all treatments except with 3-phenyl-1-propanol, and with the mixture of compounds in soil, differences of 1 and 0.5 mM are observed with 0.1 mM. For %S a significant positive correlation with concentration (*r*^2^ = 0.99) was also found when dealing with the mixture of compounds on paper (Table 2).

### 2.2. Effect of Phenolic Compounds on Lactuca sativa and Allium cepa Seedling Growth

When *p*-cresol, 2-phenylethanol and 3-phenyl-1-propanol were tested on paper separately, at concentrations above 0.5 mM, each of these compounds showed a very strong inhibition of the root and stem length of seedlings of *Lactuca sativa* and *Allium cepa* (Table 3 and Table 4). On the other hand, there was no inhibition of the development of the seedlings when the tests were conducted on commercial soil; only *p*-cresol and 2-phenylethanol at 1 mM significantly inhibited the root and cotyledon length of *A. cepa* and the root length of *L. sativa*. On the other hand, instead of having an inhibitory effect, the low concentrations of these three components even stimulated the cotyledon length growth of both species, although it should be noted that the difference was not significant.

Regarding the mixture of compounds, a significant inhibition of the size of the root and cotyledons is observed at all concentrations, in the two species and substrates tested, with some exceptions (Table 3 and Table 4). In addition, as for total germination and germination speed, the joint action of these compounds shows the same behavior in their negative action on the length of roots and cotyledons in the two species and substrates studied, showing a significantly higher inhibition than when testing with the pure compounds. Note that two exceptions are observed where we find a significantly lower inhibition, with *p*-cresol at 1 mM on the size of the root and cotyledons of *A. cepa* in soil and with 2-phenylethanol at 0.5 mM on the size of the root of *L. sativa* on paper.

When the differences between concentrations were studied, in *A. cepa* significant differences are observed in the %Root length between 1 and 0.5 mM with all the treatments studied except with 3-phenyl-1-propanol and with 2-phenylethanol in soil. Furthermore, with the mixture of compounds on paper, a significant positive correlation with concentration (*r*^2^ = 0.99) was found. For the %Cotyledon length, significant differences are also observed between 1 and 0.5 mM with all the treatments studied, but in this case, there are significant differences with 3-phenyl-1-propanol on paper and with 2-phenylethanol no differences are found neither in soil nor paper. As for the %Root length with the mixture of compounds on paper, a significant positive correlation with concentration (*r^2^* = 0.98) was found, but no significant differences between concentrations are observed in soil. For *Lactuca sativa*, significant differences are observed between 1 and 0.5 in the %Root length of all treatments except when irrigating with 2-phenylethanol on paper, and with 3-phenyl-1-propanol and with the mixture of compounds, in soil. For the %Cotyledon length on paper, significant differences of 1 and 0.5 mM are observed with 0.1 mM with *p*-cresol and 2-phenylethanol, and between 1 and 0.5 mM with 3-phenyl-1-propanol and the mixture. In soil, no significant differences are observed between the concentrations in any pure compound but in the mixture of compounds between 1 and 0.1 mM.

Furthermore, the green color was duller in the photosynthetic stems of *Allium cepa* when watered with *p*-cresol.

## 3. Discussion

Previous studies have shown that *Cistus ladanifer* is involved in the reduction of the richness and diversity of herbaceous species in Mediterranean communities where it is present [8]. This could be caused by the response to the competition for nutrients, chemical interference, the accumulation of litter, the penetration of sun light [51] and the allelopathic potential of the exudate of the leaves and photosynthetic stems of this species [8]. With the aim of helping to justify that such potential is a possible cause, the present study evaluated the phytotoxicity of *p*-cresol, 2-phenylethanol and 3-phenyl-1-propanol, which are three phenolic compounds abundant in *C. ladanifer* whose activity has not been analyzed to date [10,40,41,42,43,44,45,46].

The results of the evaluation carried out through a static acute toxicity test showed that these three compounds separately and mixing all of them have phytotoxic activity to a lesser or greater extent depending on the soil, plant species and concentration tested. In line with studies on other phenolic acids, it was observed that the three analyzed compounds had a greater inhibitory effect on root length than on stem length [18]. This effect can have negative consequences, since root growth is important for any species to establish under natural conditions.

The concentrations at which these compounds were effective are also in agreement with those reported in other studies, which show that some phenolic acids, such as cinnamic acid and ferulic acid, have the same effect on seed germination and seedling growth [18,29,35,52]. To date, the only studies found in the literature on the analyzed compounds were focused on 2-phenylethanol, and they report that this compound, through volatile emission and without being in contact with seeds of *Arabidopsis thaliana*, inhibit the germination of the latter and causes the discoloration of its cotyledons [53]. However, low concentrations of 2-phenylethanol stimulate the growth of *Aspergillus flavus*, thus revealing a hormonal behavior [54,55], trend that although the differences are not significant, we also observe the size of the cotyledons of the two species studied when the test is carried out in the soil. On the other hand, *o*-cresol has been shown to exert a negative effect on the seed germination of two types of *Brassica juncea* [56].

The tests performed on *Allium cepa*, for the three components analyzed separately and mixture at 1 mM on paper, revealed a clear inhibition of total germination, germination rate and seedling development. Similarly, the tests performed on commercial soil also showed negative effects on total germination and germination rate at all the analyzed concentrations; however, root and cotyledon length was only inhibited by *p*-cresol at 1 mM.

The results of the tests performed on *Lactuca sativa* were not as remarkable as in the case of *A. cepa*. Nevertheless, the three compounds, analyzed separately on paper at 1 mM, significantly inhibited the total germination, germination rate and seedling development of *L. sativa*. On commercial soil, no direct inhibition of germination was observed, although the germination rate was reduced, and 2-phenylethanol also inhibited the root size. This effect can have equally negative consequences. The fact that the germination process is not delayed is important for the establishment of all species under natural conditions, and root growth, since it influences the chance of the plant to settle, as it would not be able to make good use of favorable conditions, such as the first rains, or it could have difficulties to penetrate the soil. In poor ecosystems like those in which *C. ladanifer* thrives, this process can be enhanced even further, since the competition is greater, due to the lack of nutrients and the water deficit.

Since these compounds are not isolated in nature, in order to approach reality, the effect derived from the mixture of the three selected compounds was also explored. The results show that the mixture of the three compounds enhances their phytotoxic effects. It is worth highlighting that with some exceptions (a significantly lower inhibition is observed, with *p*-cresol at 1 mM on the size of the root and cotyledons of *A. cepa* in soil and with 2-phenylethanol at 0.5 mM on the size of the *L. sativa* root on paper), the separate effect of each of the three phenols on the different measured indices was lower than the effect of their combination, thus revealing that their joint presence is more effective. It is observed that in the two species tested on paper, the joint action of the three compounds at 1 mM shows a significantly greater inhibition of the measured indices than each of the compounds separately. An allelochemical may not show allelopathic activity alone, but it may show it in association with other allelochemicals [57]. The mixtures of allelochemical compounds (benzoic, *o*-coumaric, vanillic, salicylic, *p*-hydroxybenzoic and gentisic acids, scopoletin and saponarin) were more effective growth inhibitors than each individual compound (in medium and high concentrations of 0.1 and 1 mM. In these same studies, inhibition in receptor species was shown to be dose dependent and a stimulatory effect was observed with low concentrations (10^−5^ M) of benzoic, p-hydroxybenzoic and gentisic acids or a mixture of compounds [58]. Other studies with caffeic, ferulic and cinnamic acid also showed a synergistic effect [18]. Demonstrating this behavior is of great relevance, since it would imply that, in order for these compounds to be active, it may not be necessary for them to be present in high concentrations in the environment in which they exert their action. In fact, different compounds with allelopathic activity have been identified and quantified in soils, and, in most cases, their concentrations are very low, although this effect can counterweight the low concentrations present in this environment [33,59].

As in the tests conducted with the compounds separately, the joint action of the compounds in the tests performed on commercial soil showed that the effect on germination and seedling development was lower than on paper. As in other studies, Pearson’s correlation analysis showed that the concentration of mixture of phenolic compounds was correlated with the variables tested, indicating that these phytochemicals are a potential source of bioherbicides [60].

The process of how phenolic compounds reach the soil and exert their effect in it is not clear, even less so how they interact with the different variables of the soil in the different physiological functions, such as germination and seedling growth. It is known that the differences in the levels of volatile organic compounds in the soil are correlated with changes in the microbial populations of the soil [61]. Therefore, it is currently necessary to delve into the knowledge of the entire process involved in allelopathy. A factor that must be taken into account is that the compounds must reach the soil in which they exert their effect. It is important to consider that, in the tests conducted on soil, there are much more factors that influence the phytotoxic capacity of the compounds; for instance, their activity could be altered due to losses and transformations of the compounds, since they are biodegradable (e.g., *p*-cresol and 2-phenylethanol are readily biodegradable) [62,63].

However, phenolic compounds are widely known for their phytotoxic activity. The existing literature on how phenolic compounds affect the growth of receptive plants shows that these compounds cause hormonal alterations [64,65,66,67] and affect enzymatic activity [68], photosynthesis [69,70], respiration [69,71] and membrane-related processes [66,68,69,72]. However, in addition, phenolic compounds can directly inhibit germination, several studies have shown that chlorogenic acid, caffeic acid and catechol inhibit germination because they alter the functionality of the enzyme λ-phosphorylase, which has a clear relationship with seed germination [69,73]. It could be asserted that the phytotoxicity of phenolic compounds is well established and that, therefore, it can be an important source of new herbicides. These compounds could be used directly or as molecular models for the synthesis of new agrochemicals, as is the case of the monoterpene 1,8-cineol, which is present in the labdanum of *C. ladanifer* and has allelochemical properties; by modifying its molecular structure, its phytotoxicity has been increased, and it is now commercialized as Cinmethylene [74,75,76].

Currently, herbicides represent almost 50% of the pesticides used in agriculture, and their excessive use has caused environmental problems, such as soil contamination and the increase of resistant species. In view of this situation, the search for alternative herbicides to find natural components that control the undergrowth without adverse effects on the environment is fundamental. An important group of weedy species are Monocotyledoneae. Studies on biodegradable compounds with phytotoxic activity on Monocotyledoneae can contribute to this search.

These data expand our knowledge on the allelopathic potential of *Cistus ladanifer*. In conclusion, it can be asserted that *p*-cresol, 2-phenylethanol and 3-phenyl-1-pronanol, which are abundant in *Cistus ladanifer*, present phytotoxic activity, although their mere presence in the soil does not imply inhibition, since there are many factors (physical, chemical and biological) that can affect the activity of these compounds in the soil.

## 4. Materials and Methods

### 4.1. Selection of Seeds and Soils

In order to use representative models of the two taxonomic classes of plants (mono- and di-cotyledoneae), commercial seeds of *Lactuca sativa* and *Allium cepa* were selected.

Before conducting the tests, a germination test was carried out to determine the viability of the selected batches of seeds. This germination test was performed on paper and with distilled water. The results showed a total germination of over 98%.

The paper used was Whatman filter paper n° 118 (90 mm Ø). To approach reality, a commercial universal soil was used, which had the following characteristics: organic matter per dry matter (60%), electrical conductivity (mS/m), apparent dry density (320 g/L), grain size (0–20 mm) and pH 6.5.

### 4.2. Phytotoxic Activity Test

Chemically pure reagents (*p*-cresol, 2-phenylethanol and 3-phenyl-1-propanol of > 98% purity) were obtained from Aldrich-Chemical. Different solutions were prepared with MilliQ water at 1 mM, which is the maximum recommended concentration in allelopathic bioassays [73], at 0.5 and 0.1 mM of each component separately. For the mix of all three phenols, three solutions were prepared with equimolar concentrations of each of the compounds at 1 mM, 0.5 mM and 0.1 mM. To eliminate effects of pH, we measured these parameters for each solution. The pH varied between 4.9 and 5.3 from one solution to another. There were no significant differences in pH among solutions.

The bioassay conducted was a static acute toxicity test. The lettuce seeds were exposed for five days and onion seeds were exposed for six days. For each test, 50 seeds were placed in Petri dishes with 20 g of universal soil or with Whatman filter paper. Four replicates were performed, with 200 seeds in total for each solution. The Petri dishes with paper were watered with 5 mL of solution and the Petri dishes with soil were watered with 16 mL of solution. Control dishes were watered with MilliQ water. Lastly, all Petri dishes were sealed with Parafilm to prevent evaporation. The dishes were placed in a culture chamber with controlled temperature and lighting: 15 h of light and 9 h of darkness at a constant temperature of 22 °C.

### 4.3. Measured Indices to Quantify the Phytotoxic Effect

The most commonly used measures to determine the phytotoxic effects are the germination percentages and the indices that measure seedling development (total germination, germination rate, root and stem length).

It was considered that germination occurred when the testa was broken, with the subsequent emergence of the radicle, and the number of germinated seeds was counted daily for each Petri dish. With the average value of the four replicates of each treatment, total germination was calculated, as a percentage relative to the control (%TG) using the following equation:(1)$$\%\mathrm{TG}=\frac{N_{T}\cdot 100}{N}\;$$
where N_T_ is the average number of germinated seeds in each treatment and N is the average number of seeds in the control.

Germination rate (S) is one of the indices with greater sensitivity to allelopathic effects and reveals more accurately what occurs during the germination process [47,77,78]. This index was calculated using the following equation:(2)$$S=\left(N_{1}\cdot 1\right)+\left(N_{2}-N_{1}\right)\cdot \frac{1}{2}+\left(N_{3}-N_{2}\right)\cdot \frac{1}{3}+\ldots \left(N_{n}-N_{n-1}\right)\cdot \frac{1}{n}$$
where N_1_, N_2_, N_3_,…N_n−1_, N_n_ are the proportions of germinated seeds obtained in the first (1), second (2), third (3),…,(*n*−1), (*n*) days (for lettuce it was *n* = 5 and for onion *n* = 6). The results were expressed as percentages relative to the control.

The success or fitness of a seedling to establish in a certain environment is relevant to guarantee the survival of the species. The evaluation of the development of the radicle and hypocotyl is a representative indicator to determine the capacity of the plant to settle and develop. Assessing their elongation allows estimating the toxic effect of compounds that are not enough to inhibit germination but can delay or inhibit the elongation processes. Thus, the inhibition of radicle and hypocotyl elongation is a very sensitive sublethal indicator for the evaluation of biological effects on plants, providing complementary information to that obtained from the study of the effect on germination. To this end, at the end of the experiment, 10 seedlings per Petri dish were randomly selected. Their root and stem length were measured [79] and the mean value was expressed as a percentage relative to the control, thus:(3)$$\mathrm{Root}\;\mathrm{length}\;\left(\%\right)=\frac{\mathrm{treatment}\;\mathrm{root}\;\mathrm{length}}{\mathrm{control}\;\mathrm{root}\;\mathrm{length}}\cdot 100$$
(4)$$\mathrm{Cot}\mathrm{yledon}\;\mathrm{length}\;\left(\%\right)=\frac{\mathrm{treatment}\;\cot\mathrm{yledon}\;\mathrm{length}}{\mathrm{control}\;\cot\mathrm{yledon}\;\mathrm{length}}\cdot 100$$

### 4.4. Statistical Analysis

The significance level of the comparisons among treatments was estimated using the Mann–Whitney U test. The differences were considered significant when *p* < 0.05. The interrelationships between germination and seed growth with the concentration of phenolic compounds were determined by Pearson’s correlation coefficient. All statistical analyses were conducted using the statistical software SPSS 15.0.1.

## Figures and Tables

**Table 1 plants-10-01136-t001:** Effect of different concentrations of phenolic compounds from the *Cistus ladanifer* exudate on *Allium cepa* germination (%TG) and germination rate (%S), expressed as the percentage relative to the control. Four replicates of each treatment (*n* = 4 × 50 = 200 seeds in total for each solution).

	*Allium cepa*					
	%TG			%S		
Treatment	1 mM	0.5 mM	0.1 mM	1 mM	0.5 mM	0.1 mM
Paper						
*p*-cresol	70.1 * a ’	93.9 b ’	95.6 b	47.1 * a ´	57.6 * b ´	80.7 * c ´
2-phenylethanol	72.8 * a ´	81.6 * b ´	80.7 * b ´	47.4 * a ´	60.8 * b ´	53.5 * ab ´
3-phenyl-1-propanol	59.6 * a ´	75.4 * b ´	85.1 * c ´	33.1 * a ´	46.7 * b ´	54.7 * c ´
Mixture of compounds	12.5 * a	46.2 * b	98.8 c	14.5 * a	23.2 * b	90.0 c
Soil						
*p*-cresol	71.9 * a	87.6 * b ´	85.6 * b ´	55.5 * a	81.0 * b	69.2 * b
2-phenylethanol	78.7 * a	86.3 * a ´	75.3 * a	77.0 * a ´	80.1 * a	80.6 * a
3-phenyl-1-propanol	84.9 * a ´	83.5 * a ´	82.8 * a ´	73.6 * a ´	76.5 * a	70.0 * a
Mixture of compounds	73.0 * a	78.0 * a	76.7 * a	64.9 * a	79.6 * b	76.9 * b

* Significantly different from controls. ´ Significantly different from mixture of compounds. a, b, c: differences in small letters indicate significant differences between concentrations of the same index and for each treatment. *p* < 0.05 (Mann–Whitney U test).

**Table 2 plants-10-01136-t002:** Effect of different concentrations of phenolic compounds from the *Cistus ladanifer* exudate on *Lactuca sativa* germination (%TG) and germination rate (%S), expressed as the percentage relative to the control. Four replicates of each treatment (*n* = 4 × 50 = 200 seeds in total for each solution).

	*Lactuca sativa*					
	%TG			%S		
Treatment	1 mM	0.5 mM	0.1 mM	1 mM	0.5 mM	0.1 mM
Paper						
*p*-cresol	92.1 * a ´	96.2 * a ´	99.7 a	69.9 * a ´	76.0 * b ´	88.0 c ´
2-phenylethanol	96.7 * a ´	97.2 a ´	101.2 a	64.4 * a ´	86.0 * b ´	90.3 b ´
3-phenyl-1-propanol	99.2 a ´	97.2 a ´	99.2 a	80.8 * a ´	84.8 * a ´	87.9 a ´
Mixture of compounds	0.0 * a	63.9 * b	97.9 c	0.0 * a	17.0 * b	65.3 * c
Soil						
*p*-cresol	100.0 a ´	100.0 a ´	100.0 a ´	71.5 * a ´	80.9 * b ´	81.1 * b ´
2-phenylethanol	108.8 a ´	100.0 a ´	100.0 a ´	68.4 * a ´	78.5 * b ´	73.9 * ab ´
3-phenyl-1-propanol	100.0 a ´	100.0 a ´	100.0 a ´	75.7 * a ´	81.1 * a ´	82.7 * a ´
Mixture of compounds	92.0 * a	86.5 * a	92.0 a	38.6 * a	44.1 * a	59.7 * b

* Significantly different from controls. ´ Significantly different from mixture of compounds. a, b, c: differences in small letters indicate significant differences between concentrations of the same index and for each treatment. *p* < 0.05 (Mann–Whitney U test).

**Table 3 plants-10-01136-t003:** Effect of different concentrations of phenolic compounds from the *Cistus ladanifer* exudate on *Allium cepa* root length and cotyledon, expressed as the percentage relative to the control. Four replicates of each treatment (*n* = 4 × 50 = 200 seeds in total for each solution).

	*Allium cepa*					
	%Root Length			%Cotyledon Length		
Treatment	1 mM	0.5 mM	0.1 mM	1 mM	0.5 mM	0.1 mM
Paper						
*p*-cresol	51.0 * a ´	78.3 * b ´	79.4 b ´	48.5 * a ´	83.0 * b ´	92.7 c ´
2-phenylethanol	50.5 * a ´	66.0 * b ´	63.1 * b ´	71.9 * a ´	62.1 * a ´	65.5 * a
3-phenyl-1-propanol	33.0 * a ´	25.6 * a	87.5 * b ´	31.5 * a ´	51.9 * b ´	108.0 c ´
Mixture of compounds	13.5 * a	28.0 * b	47.9 * c	0.0 * a	8.5 * b	61.7 * c
Soil						
*p*-cresol	73.1 * a ´	107.0 b	107.2 b ´	77.7 * a	112.4 b ´	113.8 b ´
2-phenylethanol	108.3 a ´	110.9 a	103.7 a ´	116.6 a ´	114.2 a ´	111.3 a ´
3-phenyl-1-propanol	102.6 a ´	116.9 a ´	100.0 a ´	109.9 a ´	112.8 a ´	115.0 a ´
Mixture of compounds	89.9 * a	101.7 b	92.6 ab	80.7 * a	83.3 * a	82.4 * a

* Significantly different from controls. ´ Significantly different from mixture of compounds. a, b, c: differences in small letters indicate significant differences between concentrations of the same index and for each treatment. *p* < 0.05 (Mann–Whitney U test).

**Table 4 plants-10-01136-t004:** Effect of different concentrations of phenolic compounds from the *Cistus ladanifer* exudate on *Lactuca sativa* root length and cotyledon, expressed as the percentage relative to the control. Four replicates of each treatment (*n* = 4 × 50 = 200 seeds in total for each solution).

	*Lactuca sativa*					
	%Root Length			%Cotyledon Length		
Treatment	1 mM	0.5 mM	0.1 mM	1 mM	0.5 mM	0.1 mM
Paper						
*p*-cresol	35.4 * a ´	81.6 * b ´	93.8 c ´	82.7 * a ´	84.0 * a ´	99.1 b ´
2-phenylethanol	47.5 * a ´	48.2 * a ´	97.4 b ´	82.6 * a ´	84.0 * a ´	99.1 b ´
3-phenyl-1-propanol	83.6 * a ´	95.8 b ´	108.2 b ´	88.0 * a ´	112.5 b ´	102.6 b ´
Mixture of compounds	0.0 * a	64.2 * b	119.7 * c	0.0 * a	8.5 * b	61.7 * c
Soil						
*p*-cresol	93.5 a	105.1 b	102.2 b	111.1 a ´	111.9 a ´	111.7 a ´
2-phenylethanol	86.6 * a	96.5 b	90.9 b	115.2 a ´	104.3 a ´	112.4 a ´
3-phenyl-1-propanol	95.7 a	104.3 a	94.5 a	109.3 a ´	111.9 a ´	111.7 a ´
Mixture of compounds	90.7 * a	92.7 a	101.3 a	87.5 * a	94.4 ab	97.1 b

* Significantly different from controls. ´ Significantly different from mixture of compounds. a, b, c: differences in small letters indicate significant differences between concentrations of the same index and for each treatment. *p* < 0.05 (Mann–Whitney U test).

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
