# Peer review of "Phytotoxic Activity of *p*-Cresol, 2-Phenylethanol and 3-Phenyl-1-Propanol, Phenolic Compounds Present in *Cistus ladanifer* L."

_plants, 2021, doi:10.3390/plants10061136_

Round 1

Reviewer 1 Report

The article entitled "Phytotoxic activity of p-cresol, 2-phenylethanol and 3-phenyl-2 1-propanol, phenolic compounds present in Cistus Ladanifer L. " reports the phytotoxitc activity of 3 phenolic compounds (p-cresol, 2-phenylethanol and 3-phenyl-1-propanol ) present in the  labdanum of C. ladanifer. This phytotoxic activity has been monitored on the germination, the germination rate, the root length and the cotyledon length of 2 plants (Allium cepa and Lactuca sativa). This article is well written with clear conclusions and it can be accepted just with minor corrections.

minor comments:

  • For each table, add the number of replicates and the standard deviations
  • For the materials and methods parts:
    • add explanation for the choice of Allium cepa.
    • Indicate how the pH has been adjusted after application of the different compounds
  • line 14: complete the sentence: “… have been evaluated” for ?
  • line 252 : indicate the purity of the compounds
  • line 289 : translate in English

Author Response

A1. We thank the reviewer for his/her vote of confidence. We have addressed below all the comments raised.

R1.1 For each table, add the number of replicates and the standard deviations

A1.1Following the comment by the reviewer, we have now included the number of replicates in the revised manuscript and other statistical analyzes

R1.2 For the materials and methods parts:

  • add explanation for the choice of Allium cepa.

A1.2 In order to address the Referee’s comment, we have included the following paragraph in the revised manuscript: (lines 100 - 108):

In order to use representative models of the two taxonomic classes of plants (mono- and di-cotyledoneae), commercial seeds of Lactuca sativa and Allium cepa were selected. These species are ideal for adequately showing the allelochemical effects on the germination processes [18, 47]. Moreover, L. sativa is recommended by the US EPA (United States Environmental Protection Agency) for phytotoxicity tests, being among the most sensitive species [48]. Allium cepa has proved to be the most useful, and has repeatedly been suggested as a standard test material by International Organization for Standarization [49], and the Organization of Economical Co-operation and Development the citation in the list of species historically used in plant testing [50].

R1.3

  • Indicate how the pH has been adjusted after application of the different compounds

A1.3 In order to clarify this point, we have included the following sentence in the revised manuscript (lines 466 - 470):

For the mix of all three phenols, three solutions were prepared with equimolar concentrations of each of the compounds at 1mM, 0.5mM and 0.1mM. To eliminate effects of pH, we measured these parameters for each solution. The pH varied between 4.9 and 5.3 from one solution to another. There were no significant differences in pH among solutions.

R1.4

  • line 14: complete the sentence: “… have been evaluated” for ?

A1.4 This sentence is developed or explained in lines 54-60. The list of evaluated compounds can be seen in reference 11 cited in line 61 (Chaves, N.; Alías, J.C.; Sosa, T. Phytotoxicity of Cistus ladanifer L.: Role of allelopathy. Allelopath. J. 2016, 38(2), 113–132)

R1.5

  • line 252 : indicate the purity of the compounds

A1.5 We have now included the purity of the compounds (line 463)

R1.6

  • line 289: translate in English

A1.6 we have included the translate in English

Reviewer 2 Report

The submitted manuscript presents a very interesting research topic, however, in my opinion this potential has not been fully exploited.  The reason for my concern is that a simplified statistical approach was used to analyze the data.

I suggest the use of such statistical tools that show not only differences in the performance of individual treatments relative to controls, but also differences among all treatments, i.e., variants of the experimental factors (e.g. type of allelocompounds, allelocompound doses, type of medium). Please consider using, for example, analysis of variance along with the appropriate tests assessing differences between objects. When presenting data in tables, a common way to show differences between objects is to mark them with different letters.

Once the statistical analysis is extended, a more thorough discussion of the results will be needed involving its use. A synergistic effect of the three compounds as well as the stimulation effect of some of them must be statistically proved.

The study seems to have shown a hormetic effect (possibly significant) of the allelochemicals tested. It is worth taking a closer look at this issue as well. 

Other specific comments are provided in the manuscript attached.

I hope my suggestions will help Authors improve the article in terms of content and form.

Author Response

R2.1 The submitted manuscript presents a very interesting research topic, however, in my opinion this potential has not been fully exploited.  The reason for my concern is that a simplified statistical approach was used to analyze the data. I suggest the use of such statistical tools that show not only differences in the performance of individual treatments relative to controls, but also differences among all treatments, i.e., variants of the experimental factors (e.g. type of allelocompounds, allelocompound doses, type of medium). When presenting data in tables, a common way to show differences between objects is to mark them with different letters

A2.1 The Referee is right and we corrected the revised manuscript accordingly (see Tables 1-4). The statistical analyzes carried out to evaluate possible differences between controls showed significant differences between the different variables measured from the controls of the different substrates and between the controls of the species used. For this reason, differences in type of medium and species was not considered to show this type of analysis in the paper.

R2.2 Please consider using, for example, analysis of variance along with the appropriate tests assessing differences between objects.

A2.2 Analysis of variance is not considered as it is non-parametric data.

R2.3 Once the statistical analysis is extended, a more thorough discussion of the results will be needed involving its use.

A2.3 The results and discussion was completed. See the manuscript attached.

R2.4 A synergistic effect of the three compounds as well as the stimulation effect of some of them must be statistically proved. The study seems to have shown a hormetic effect (possibly significant) of the allelochemicals tested. It is worth taking a closer look at this issue as well. 

A2.4 The Referee is right, after analyzing the data, it is found that it cannot be declared that there is a synergistic effect when the compounds act together, nor can we speak of stimulation and hormetic effect because there are no significant differences between the concentrations involved.

R2.5 Other specific comments are provided in the manuscript attached.

A2.5 Other specific comments were repaired. See attached manuscript.

R2.6 I hope my suggestions will help Authors improve the article in terms of content and form.

A2.6 We thank the reviewer for his/her vote of confidence. We have addressed all the comments raised.

Reviewer 3 Report

Table 1

Replace “mix” with “mixture of”

Table 1

In most of the cases, TG% and GR% of seed in the presence of phenolic compounds at 0.5 mM was higher than phenolic compounds at 0.1 mM. Explain with giving recent citations.

Table 1: Explain why mixture of phenolic compounds showed lower TG% and GR% of seed compare to individual compound. Give supporting citations.

Table 2:

TG% and GR% of seed at conc. 0f 1 mM mixture phenolic compounds was zero. Discuss with giving possible reasons in the text.

Line 129-130: “These indices also showed a correlation between the concentration and the effect”. Where is the correlation data? Find the correlation between the parameters and discuss in the text.

Authors of this manuscript can try assessing the allelochemical test of Cistus Ladanifer L. using plant extracts.

Author Response

A2.We thank the reviewer for his/her vote of confidence. We have addressed below all the comments raised.

R2.1 Table 1

Replace “mix” with “mixture of”

A2.1 mix was replaced by mixture

R2.2 Table 1 In most of the cases, TG% and GR% of seed in the presence of phenolic compounds at 0.5 mM was higher than phenolic compounds at 0.1 mM. Explain with giving recent citations.

A2.2 After analyzing the data, it is found that it cannot be declared that there is a synergistic effect when the compounds act together, nor can we speak of stimulation and hormetic effect because there are no significant differences between the concentrations involved.

In order to address the Referee’s comment, we have included the following paragraph in the revised manuscript (lines 381-387): The mixtures of allelochemical compounds (benzoic, o-coumaric, vanillic, salicylic, p-hydroxybenzoic and gentisic acids, scopoletin and saponarin) were more effective growth inhibitors than each individual compound (in medium and high concentrations of 10−4 and 10−3M. In these same studies, inhibition in receptor species was shown to be dose dependent and a stimulatory effect was observed with low concentrations (10-5 M) of benzoic, p-hydroxybenzoic and gentisic acids or a mixture of compounds [58].

R2.3 Table 1: Explain why mixture of phenolic compounds showed lower TG% and GR% of seed compare to individual compound. Give supporting citations.

A2.3 In order to clarify this point, we have included the following sentence in the revised manuscript (lines 362 - 384): It is observed that in the two species tested on paper, the joint action of the three compounds at 1mM shows a significantly greater inhibition of the measured indices than each of the compounds separately. An allelochemical may not show allelopathic activity alone, but it may show it in association with other allelochemicals [57]. The mixtures of allelochemical compounds (benzoic, o-coumaric, vanillic, salicylic, p-hydroxybenzoic and gentisic acids, scopoletin and saponarin) were more effective growth inhibitors than each individual compound (in medium and high concentrations of 10−4 and 10−3M.

R2.4 Table 2: TG% and GR% of seed at conc. 0f 1 mM mixture phenolic compounds was zero. Discuss with giving possible reasons in the text.

A2.4 In order to address the Referee’s comment, we have included the following paragraph in the revised manuscript (lines 415-418): But in addition, phenolic compounds can directly inhibit germination, several studies have shown that chlorogenic acid, caffeic acid and catechol inhibit germination because they alter the functionality of the enzyme λ-phosphorylase, which has a clear relationship with seed germination [69,73].

R2.5 Line 129-130: “These indices also showed a correlation between the concentration and the effect”. Where is the correlation data? Find the correlation between the parameters and discuss in the text.

A2.5 We have now included a further comparison: a correlation between the concentration and the effect (see Tables 1-4). The interrelationships between germination and seed growth with the concentration of phenolic compounds were determined by Pearson's correlation coefficient.

R2.6 Authors of this manuscript can try assessing the allelochemical test of Cistus ladanifer L. using plant extracts.

A2.6These tests have already been done in previous studies. See review: Chaves, N.; Alías, J.C.; Sosa, T. Phytotoxicity of Cistus ladanifer L.: Role of allelopathy. Allelopath. J. 2016, 38(2), 113–132

Round 2

Reviewer 2 Report

The manuscript was improved. Thank you to the Authors for the thorough revision.

I do have some minor comments.

  • I would still suggest making some corrections to the abstract. Please avoid rather the use of the expressions such as: “the results show” or “it is observed that”. Using short and explicit conclusions is usually suggested. Additionally, practical recommendations are also welcome.
  • What is adopted as a "treatment"? Are p-cresol, 2-phenylethanol and 3-phenyl-1-propanol, and mixture of compounds the treatments? If yes, I believe that the word "treatment" should be introduced in the head of each table (column 1).

I would like to wish the Authors success in their further scientific work.

Author Response

Following the comment by the reviewer, we have now rephrased the paragraph of the Abstract.

For each table was added the word "treatment"

We thank the reviewer for all his contributions that undoubtedly help us to improve.

Best regards

Reviewer 3 Report

The authors of the manuscript have revised the manuscript properly, addressed all the issues raised during the first round of reviewing process.

I think, the manuscript can be accepted and suitable for publication in the Plants journal. BEST OF LUCK

Author Response

We thank the reviewer for all his contributions that undoubtedly help us to improve.

Best regards